# The Role of EUS-Guided FNA and FNB in Autoimmune Pancreatitis

**DOI:** 10.3390/diagnostics11091653

**Published:** 2021-09-09

**Authors:** Nicolò de Pretis, Stefano Francesco Crinò, Luca Frulloni

**Affiliations:** Department of Medicine, Gastroenterology Unit, Pancreas Center, University of Verona, 37134 Verona, Italy; stefanofrancesco.crino@aovr.veneto.it (S.F.C.); luca.frulloni@univr.it (L.F.)

**Keywords:** autoimmune pancreatitis (AIP), diagnosis, EUS-guided fine-needle aspiration (FNA), EUS-guided fine-needle biopsy (FNB), histology

## Abstract

Autoimmune pancreatitis (AIP) is an increasingly recognized disease classified into two different subtypes based on histology. According to the International Diagnostic Criteria (ICDC), the diagnosis is achieved using a combination of different criteria. In patients presenting with a typical imaging appearance, the diagnosis may be straightforward, and steroid treatment is recommended, even without histological confirmation. In patients with atypical imaging or mass-forming appearance, the differential diagnosis with pancreatic cancer is challenging and crucial for treatment strategy. Endoscopic ultrasound (EUS)-guided tissue acquisition has been proposed to achieve a histological diagnosis. Fine-needle aspiration (FNA) was first proposed to aspirate cells from pancreatic lesions. Despite excellent results in terms of sensitivity for pancreatic cancer, the data are disappointing regarding the diagnosis of AIP. The recent development of new needles allowing fine-needle biopsy (FNB) has been associated with improved diagnostic accuracy based on preserving the tissue architecture, which is necessary to detect the typical histological features of AIP. However, the published literature on the role of EUS-guided FNA and FNB is limited and mainly focused on type 1 AIP. The present study aimed to review the available literature on the role of EUS-guided FNA and FNB in the diagnosis of AIP.

## 1. Introduction

Autoimmune pancreatitis (AIP) is a fibroinflammatory disease of the pancreatic gland with a postulated autoimmune mechanism [1]. Based on the International Consensus Diagnostic Criteria (ICDC), the term AIP unifies two different diseases with distinct histological features but similar imaging appearance, type 1 and type 2 AIP [2]. Type 1 is considered part of IgG4-related disease [3] and is histologically characterized by increased IgG4-positive cells, obliterative phlebitis, storiform fibrosis, and periductal lymphoplasmacytic infiltrates. It may be associated with other organ involvement and elevated serum IgG4 levels. Type 2 may be associated with ulcerative colitis and is histologically characterized by granulocytic epithelial lesions (GELs), sometimes with granulocytic acinar inflammation. Patients without definitive criteria for the differentiation between types 1 and 2 are classified as not otherwise specified (NOS) [4].

Type 1 and type 2 AIP are not distinguishable based on imaging. In both subtypes, the inflammatory process may involve the pancreas diffusely or focally.

Diffuse forms have typical imaging appearances characterized by enlargement of the whole pancreas, hypoenhancement in the early arterial phase and slight hyperenhancement in the late phase. Additionally, a hypoenhanced peripheral capsule-like rim in the arterial phase is frequently detected, and the main pancreatic duct may be compressed (long strictures involving more than 1/3 of the length) without or with mild upstream dilation. These features do not raise suspicion for pancreatic cancer, and the diagnosis of AIP is simpler, particularly in the presence of other diagnostic criteria, such as serum IgG4 elevation or other organ involvement.

By contrast, focal forms are a real clinical challenge because imaging appearance may be confused with pancreatic cancer. In these cases, representing approximately 50–60% of patients, AIP appears as a mass-forming lesion with frequent concomitant main pancreatic duct dilation and common bile duct obstruction. In the absence of other concomitant diagnostic criteria, the risk of misdiagnosing AIP as pancreatic cancer is high. Therefore, a significant proportion of focal AIP patients undergo inappropriate surgical resection even at experienced centers.

Tissue acquisition for histological diagnosis should ideally be the gold standard for the diagnosis of AIP, differential diagnosis with pancreatic cancer and differentiation between type 1 and type 2 AIP. Based on ICDC, only tissue specimens obtained by core biopsy and surgical resection are suitable to diagnose AIP histopathologically, but a clear definition of “core biopsy” is lacking [5].

According to the ICDC, typical histological features for type 1 AIP include the following [2,6]:

-Lymphoplasmacytic infiltration: this feature may be observed in and around the pancreatic parenchyma; concomitant fibrosis is frequently observed.

-IgG4-positive plasma cells at frequently >10/high-power fields.

-Storiform fibrosis is generally characterized by spindle-shaped cells and inflammatory cells in the background of delicate collagen.

-Obliterative phlebitis: an obstructive or stenotic venous lesion with characteristic inflammatory changes.

The presence of three or more of these features is considered a level 1 histologic criterion for type 1 AIP. The presence of two of these features is considered level 2. A definitive diagnosis of type 1 AIP based on histology is achieved only with level 1 histology.

According to the ICDC, some typical histological features of type 2 AIP include the following [2,6]:

-The presence of GELs is considered a level 1 histologic criterion.

-Neutrophilic infiltration in the lobules was considered a level 2 histologic criterion.

A definitive diagnosis of type 2 AIP based on histology is achieved only with level 1 histology.

This paper aimed to review the available literature on the role of endoscopic ultrasound (EUS)-guided tissue acquisition (FNA and FNB) in patients with AIP.

## 2. Fine-Needle Aspiration (FNA) and Fine-Needle Biopsy (FNB)

Endoscopic ultrasound is an excellent technique for the morphologic evaluation of the pancreas, mainly characterized by altered echotexture, reduced echogenicity and hyperechoic strands, but in focal forms of AIP, has non-specific appearance. Therefore, tissue acquisition is required for a definitive diagnosis.

The aim of FNA is the aspiration of cells obtained from a target tissue using a conventional straight needle. This technique allows the pathologist to evaluate the presence and features of abnormal cells in the aspirated sample and sometimes of small bioptic samples. Specifically designed needles have been developed to obtain an intact histologic core sample (fine-needle biopsy, FNB) with preserved tissue architecture (Figure 1).

### 2.1. FNA in AIP

Considering the rarity of the disease, all the published studies on FNA and AIP include small sample sizes and are focused on type 1 AIP. Very few data have been published on type 2 AIP.

#### 2.1.1. FNA for the Differential Diagnosis between AIP and Pancreatic Cancer

Most of the studies have focused on the differential diagnosis of AIP and pancreatic cancer, which is frequently the main issue in clinical practice, considering prognostic implications and different therapeutic approaches.

Imai and colleagues reported a retrospective series of 85 FNAs using 22-gauge needles (3 to 5 passes) on patients with pancreatic masses greater than 2 cm [7]. The sensitivity for malignancy was 92.2% in the 64 patients with pancreatic cancer. In the 21 cases of AIP, none were inadequate, and none were misdiagnosed with cancer. However, the typical histological appearance of AIP (plasma cell infiltration, IgG4-positive plasma cells, obliterative phlebitis, storiform fibrosis, or the presence of GELs) has not been detected in any patient. Therefore, the diagnosis of AIP was achieved based on non-histological criteria (imaging, other organ involvement, serum IgG4, and response to steroids).

Based on these data, some authors have proposed FNA to rule out malignancy. A retrospective study on FNA (18-, 22-, and 25-gauge needles) in 40 AIP patients reported a histological diagnosis of AIP only in six patients (15%), with a single patient (2.5%) misdiagnosed with acinar cell carcinoma [8]. Among the remaining AIP patients without a histological diagnosis, the authors found significant differences in the radiological features and Ca19.9 levels compared with those with pancreatic cancer and inconclusive FNA. Therefore, considering that the histological diagnosis of AIP by FNA is difficult, the authors concluded that FNA may be used to rule out malignancy in patients with AIP.

The detection of DNA mutations in FNA tissue has been proposed to improve the discrimination between AIP and pancreatic cancer [9]. Khalid and colleagues published a retrospective study comparing 16 patients with pancreatic cancer and 16 patients with AIP. DNA amplification was possible only in 11 patients with pancreatic cancer and 15 with AIP. Among the 11 patients with pancreatic cancer, 5 FNAs were considered positive for malignancy, 4 suspicious for malignancy, and 2 inconclusive. In the AIP group, the number of inconclusive FNAs was higher (10/15). Only 3 of 15 were consistent with an inflammatory process, and 2 were considered suspicious for malignancy. Despite the discouraging results of FNA, after DNA amplification, *KRAS* mutation was detected in none of the AIP cases but in 10/11 of the pancreatic cancer cases. The authors concluded that a *KRAS* mutation in FNA material from a pancreatic mass may help discriminate pancreatic cancer from inflammatory conditions such as AIP. Despite the small sample size and lack of needle diameter data, this paper has the merit of including a real challenging population, namely patients with suspected pancreatic cancer.

#### 2.1.2. FNA for the Diagnosis AIP

The role of FNA in achieving an AIP diagnosis has been investigated in few studies.

The largest published paper [10] is a prospective multicenter study including 78 AIP patients with parenchymal and ductal imaging features suggesting AIP according to ICDC. Additionally, more than 80% of patients had serum IgG4 elevation (>135 mg/dL), and more than 55% had other organ involvement. A 22-gauge FNA needle was used (both the slow-pull and aspiration methods) with a mean number of punctures of 3.4 ± 1.3. Despite a highly selected population with no patients with mass-forming lesions or suspected cancer, only 32 patients (41%) had a definitive histological diagnosis of type 1 AIP (a level 1 histology criterion based on ICDC). In an additional 13 patients (17%), a level 2 histological diagnosis was achieved. Surprisingly, among the 25 patients with a definitive diagnosis of type 1 AIP based on clinical, radiological, and serological criteria (without histology), 11 (44%) had a nondiagnostic FNA.

The same author previously published a smaller series of 25 AIP patients in 2012 [11], reporting a higher rate of hydrological diagnosis (56% level 1 histology, 24% level 2 histology and only 20% nondiagnostic FNA) with a 22-gauge FNA needle. No data were available on the number of passes or aspiration technique.

Another relatively large multicenter prospective study, again from Japan, collected 50 patients with suspected AIP [12]. Twenty-seven patients (54%) already had a definitive diagnosis of type 1 AIP based on clinical, radiological, and serological criteria. FNA was performed with a 22-gauge needle with a mean number of punctures of 2.02 ± 0.48. None of the patients met the level 1 histological criterion for type 1 AIP, and 68% met the level 2 criterion. Moreover, in three cases, GELs were detected, and a diagnosis of type 2 AIP was achieved. The authors reported that the addition of pathological evaluation improved the diagnostic accuracy in just 8 out of the 50 patients (16%) and concluded that EUS-FNA with a 22-gauge needle is not an effective diagnostic method for most patients with AIP.

Cao and colleagues published a single center prospective series of EUS-guided FNA with a 22-gauge needle on 27 patients with imaging suggestive of AIP [13]. Despite the highly selected population without challenging mass-forming lesions, levels 1 and 2 histological criteria according to the ICDC were reported only in 5 (19%) and 12 (44%) patients, respectively. Similar data have been reported by another Japanese study on 47 AIP patients undergoing EUS-guided FNA with a 22-gauge needle [14]. In this retrospective study, the authors found that just nine (19%) patients met the level 1 histological finding for type 1 AIP, and five patients met the level 2 findings (11%). Although no GELs were detected (level 1 histological finding), three patients (6%) had level 2 findings (granulocytic acinar infiltrate) for type 2 AIP.

Table 1 reports the main results of the studies evaluating the role of EUS-guided FNA using a 22-gauge needle to diagnose type 1 AIP.

The diagnostic accuracy of EUS-guided FNA was not more satisfactory using larger needles. In a retrospective study involving 44 patients with AIP, the authors reported only 17 samples (39%) diagnostic for AIP using a 19-gauge FNA needle [15]. Recently, Sugimoto and colleagues showed more promising results using a 19-gauge FNA needle and a special sampling technique (“wet suction technique”) [16]. In detail, the authors described 11 patients with suspected type 1 AIP, reporting level 1 histological findings in 4 (36.4%) and level 2 histological findings in the other four patients (36.4%).

No data are available on rare type 2 AIP cases included in some studies with type 1 AIP or in case reports [17].

### 2.2. FNB in AIP

To overcome the diagnostic limitations of FNA, biopsy needles have been developed, allowing not only a cytological evaluation but also histologic examination by preserving the tissue architecture (Figure 2).

The first published reports were on EUS-guided core pancreatic biopsies using tru-cut biopsy needles. In 2005, Levy and colleagues described three patients with obstructive jaundice and suspected AIP in whom malignant obstruction could not be excluded and had undergone EUS-guided tru-cut core biopsy. In two patients, the diagnosis of AIP was achieved; in the third, only non-specific chronic inflammation was described [18]. Some years later, a retrospective series from Japan including 14 patients was published. Patients had undergone both EUS-guided FNA and EUS-guided tru-cut core biopsy for suspected AIP [19]. In the eight patients with an established clinical diagnosis of AIP, histological diagnosis was achieved only in 3 (37%) with FNA and in all (100%) with tru-cut core biopsy. These studies suggest that the preservation of tissue architecture may increase the diagnostic accuracy of tissue acquisition methods for AIP. A German group published in the same year a retrospective series of 26 patients with AIP who had undergone pancreatic tru-cut core needle biopsies [20]. Unfortunately, most of the biopsies were performed using transabdominal ultrasound guidance; only three were performed using EUS guidance. No specific data are available on EUS-guided procedures. However, the authors reported a global sensitivity for AIP of 86% and presented the first data on tru-cut core biopsies and type 2 AIP (14 biopsies showed the presence of GELs). Some years later, the first case series on EUS-guided core biopsies showing type 2 AIP was published in a paediatric setting. Nine patients with a mean age of 13.6 years had undergone EUS-guided tru-cut core biopsy for pancreatitis or other pancreatic or biliary symptoms; in six (67%) patients, a histologic diagnosis of type 2 AIP was achieved [21].

However, tru-cut core biopsy needles have no longer become available because of scarce maneuverability. The first second-generation FNB needle was the side-fenestrated reverse-beveled needle. Unfortunately, disappointing results have been reported, with no substantial advantage compared with standard FNA needles [22]. More recently, newly designed FNB forward-acquiring needles have been developed, including the following:(a)Franseen-tip needles;(b)Side-fenestrated forward-cutting beveled needle (20-gauge caliber available only);(c)Fork-tip needles;(d)Menghini-type needles.

Histologic and diagnostic yields obtained using these new-generation needles outperformed both standard FNA [23,24] and reverse-beveled side-fenestrated needles [25] to evaluate solid pancreatic lesions. Therefore, EUS-guided sampling is moving from FNA to FNB [26].

The role of EUS-guided FNB in AIP patients was first investigated in a recent prospective, randomized, controlled, multicenter study comparing a 22-gauge forward-acquiring needle with a Franseen-like crown and a 20-gauge forward-cutting beveled needle in suspected type 1 AIP patients [27]. One hundred ten consecutive patients with suspected type 1 AIP were included, but nine patients were excluded from the analysis because the histological diagnosis was different from that of type 1 AIP. No tissue specimens were obtained in 6 of 50 patients in the Franseen group (12%) or in 12 of 51 (25%) in the forward-bevel group (*p* = 0.19). The presence of level 1 histologic criteria according to ICDC was diagnosed in 56% and 26% of the Franseen group and forward-beveled group, respectively (*p* = 0.001). The authors concluded that the biopsy tissue obtained using the 22-gauge Franseen needle provided a more accurate diagnosis of AIP than that obtained using a 20-gauge forward-bevel needle (sensitivity: 78% vs. 45%). Similar results have been published in a prospective multicenter study by Ishikawa and colleagues [28]. Fifty-six patients with suspected AIP (pre-EUS Level 1 or 2 imaging appearance based on ICDC) had undergone EUS-guided FNB using a 22-gauge Franseen needle with average passes of 2. Forty-two patients (76%) already had a definitive diagnosis of type 1 AIP based on serology and imaging. Among the remaining 13 patients, 8 achieved a definitive diagnosis of type 1 AIP after FNB. Finally, 8 of 56 (14%) gained a real clinical advantage from the procedure. However, the authors reported level 1 histology for AIP in 58% and level 1 or 2 histology in 93%. Moreover, if the length of core tissue was >10 mm, the presence of level 1 or 2 histology was 100%; if the length was >20 mm, the presence of level 1 rose to 86%. These data suggest that endoscopists should focus on the FNB technique to obtain an appropriate percentage of “long” core tissues that may improve the diagnostic accuracy in AIP [29].

These two studies have several strengths, such as the prospective design and large sample size. Both presented encouraging results of FNB as a diagnostic tool in AIP patients, suggesting using a 22-gauge Franseen needle in patients with suspected AIP. However, both studies have some limitations. First, the included population was highly selected (pre-EUS suspicion of type 1 AIP). Therefore, challenging patients with an atypical imaging appearance, mass-forming lesions and potential misdiagnosis with pancreatic cancer were not included. Additionally, the clinical advantage of patients already diagnosed with definitive type 1 AIP before EUS-guided FNB is debatable [30].

This finding was confirmed by another retrospective study from Japan evaluating the efficacy of EUS-guided FNB (with different needle types and diameters) in type 1 AIP patients [31]. Among 85 neoplastic pancreatic lesions, 28 presented as diffuse enlargement of the pancreatic gland, and 57 presented as focal/segmental swelling. Based on the ICDC, only 22 cases (26%) were diagnosed as histologic level 1, and 23 (27%) were diagnosed as level 2. In 40 patients (47%), FNB was useless. The more limited results of FNB in this study may be related to the involved population, considering that the authors included not only typical AIP patients but also a significant proportion of mass-forming lesions. However, FNB has been essential for the diagnosis of type 1 AIP in 33% of patients with segmental/focal pancreatic enlargement, while it contributes to the diagnosis in only 4% of patients with diffuse pancreatic swelling, confirming that, in such a population, FNB may not be necessary. However, the lack of uniformity of needle type and diameter is a main limitation in this paper [32].

The only paper including EUS-guided FNB performed using Fork-tip type needles was published by a British group, reporting a retrospective series of 24 patients with a final diagnosis of AIP [23]. Interestingly, a significant proportion of cases had radiological features concerning cancer—namely focal mass of the pancreas (33%) and biliary obstruction (79%). Six patients had undergone EUS-guided FNB using a reverse-bevel needle without obtaining diagnostic specimens. Eighteen patients had undergone FNB using a fork-tip needle, achieving histological level 1 in 72% and level 2 in 7%; 22% were not diagnostic. These data confirm that side-bevel needles should probably not be recommended as a first-line approach in EUS-guided sampling of suspected AIP [33].

Finally, a recent paper from Japan reported a retrospective series of EUS-guided FNB procedures using a new needle type, originally developed for liver biopsies (Menghini-type needle), on 14 AIP patients [34]. The tapered beveled edge of the outer needle and inner needle connected to a barrel equipped with an aspiration piston should improve the aspiration of tissue, reducing blood contamination. The authors reported level 1 histology in 5 patients and level 2 histology in 4 patients. Table 2 reports data from the studies focused on the role of EUS-guided FNB in the type 1 AIP diagnosis.

Finally, few case reports have been published on the role of EUS-guided FNB in the diagnosis of type 2 AIP, reporting some cases of GEL identification in tissue specimens of Fork-Tip needles [35]. Table 3 reports the few studies reporting data on EUS-guided FNA and FNB in type 2 AIP.

## 3. Complications

Despite the large number of patients included in all the reported studies, a low rate of adverse events has been described in both FNA and FNB. Ishikawa et al. [28] reported the highest published complication rate in FNB, with two patients with adverse events out of a population of 56 patients (3.6%): two cases had mild abdominal pain, with one requiring overnight hospitalization. Kurita et al. reported two adverse events out of 101 patients (2.0%): one case of acute pancreatitis and one case of mild self-limiting bleeding [27].

The adverse event rate was higher in studies on EUS-guided tru-cut core biopsies (33–44%) [18,21] and on Menghini-type needles (14%) [31].

No significant or lethal complications were described.

## 4. Discussion

Despite technological improvements, the diagnosis of AIP remains difficult in clinical practice if typical radiological and serological criteria are lacking. Almost all available literature on the role of EUS-guided FNA or FNB is focused on type 1 AIP, while the published data on type 2 AIP are anecdotal. This finding is likely related to the rarity of type 2 AIP, particularly in Eastern countries, which largely contributed to the current knowledge on this topic. Very few papers have been published from European or American groups.

Clinical, radiological, and serological features allow the achievement of a definitive diagnosis of AIP in many patients. In the case of atypical mass-forming imaging and a lack of other diagnostic criteria, tissue sampling is mandatory to make a diagnosis of AIP and to exclude pancreatic cancer. According to the ICDC, some histological features have been established as typical for type 1 and type 2 AIP.

Storiform fibrosis and obliterative phlebitis appear to be those more rarely identified, not only on pancreatic specimens as evidenced in the present review but also in biopsies performed in other organs involved in IgG4-related disease [36].

The first experience with pancreatic sampling was EUS-guided FNA. The results of FNA for the diagnosis of pancreatic cancer are excellent. By contrast, the results are disappointing in diagnosing AIP, with level 1 histology achieved between 0% and 56%. Additionally, almost all studies involved patients with a typical imaging appearance and a definitive diagnosis of AIP already achieved based on the other diagnostic criteria. Therefore, some authors proposed FNA to exclude pancreatic cancer more than for a definitive histological diagnosis of AIP. Based on the available literature, this approach appears to be reasonable. The response to steroids is a cardinal criterion based on ICDC, particularly in patients without typical radiological and serological criteria. The exclusion of pancreatic cancer by FNA negative for malignancy may be critical for clinical management, allowing the administration of steroids, which may be necessary to achieve the diagnosis of AIP.

Some promising data have been published on tru-cut core biopsies, but this type of needle is not widely available.

In recent years, new FNB needles have been developed to maintain tissue architecture in samples [37]. These needles showed better results in achieving a histological diagnosis of AIP than FNA [38]. Side-fenestrated needles appear to be less accurate in the diagnosis compared with front-cutting needles, such as Franseen and Fork-tip needles, which showed level 1 histology in approximately 56–72% of cases. Additionally, some data suggest that larger pancreatic fragments may impact the diagnostic accuracy of AIP. Therefore, endoscopists should focus on improving their technique to obtain larger tissue fragments.

A recently published meta-analysis included 440 patients with AIP, 309 of whom had undergone EUS-guided FNA and 131 who had undergone EUS-guided FNB [39]. The pooled diagnostic yields for level 1 or 2 histology criteria of AIP were 55.8% for FNA and 87.2% for FNB (*p* = 0.03). Additionally, the authors reported a 30.0% rate for level 1 histology with FNA and a 60.1% rate with FNB sampling (*p* < 0.01). The better outcomes of FNB than FNA in patients with AIP have been confirmed in a recent systematic review [40]. Three hundred seventy-six patients with AIP who had undergone EUS-guided FNA and 196 patients who had undergone EUS-guided FNB were included, reporting diagnostic sensitivities of 42.0% and 60.2%, respectively (*p* > 0.01). However, both papers [39,40] included tru-cut core biopsies in the FNB group, which may have influenced the results significantly.

Despite the diagnostic superiority of FNB needles compared with FNA needles, the diagnostic accuracy is still not excellent. The largest part of the studies was conducted on selected patients, frequently with a definitive AIP diagnosis already based on radiological and serological criteria. Furthermore, some patients with a definitive diagnosis of AIP based on imaging and serology have an inconclusive histological diagnosis after FNA or FNB. These findings may have some implications in clinical practice, particularly in the case of reconsidering a definitive diagnosis of AIP after nondiagnostic tissue sampling.

Notably, all the published studies referred to ICDC histological criteria for the diagnosis of AIP. However, these criteria are clearly recognized only on surgical specimens and tru-cut biopsies, which are currently rarely performed. Consequently, diagnostic criteria should be tailored to tissue fragments that can be collected with EUS-FNB, representing the standard of care for preoperative pancreatic tissue sampling. Additionally, the diagnostic findings of AIP on biopsy specimens, as well as immunohistochemical staining, should be standardized. With this purpose, Notohara et al. published a pathologists’ guidance document to diagnose AIP using biopsy tissues [6]. Although this paper does not properly represent a guideline, it helps standardize EUS-FNB sample processing and interpretation.

New pancreatic biopsy techniques, such as a through-the-needle biopsy using microforceps, have been proposed, but the data are very limited [41].

## 5. Conclusions

The new FNB needles appear to be more accurate in diagnosing AIP than FNA needles, and their use should be limited to exclude cancer in selected patients. EUS-guided FNB should be considered a first-line approach for pancreatic sampling. Franseen-type and Fork-tip-type needles are superior to side-fenestrated needles and should be preferred. The aspiration of large fragments (>10 mm) may improve the diagnostic accuracy for AIP. Tissue sampling should probably not be performed if a definitive diagnosis of AIP is already achieved based on imaging and serology. Finally, no definitive data have been published on FNB in patients with AIP, atypical imaging, and serum IgG4 negativity. Future large prospective multicenter studies are needed to evaluate the diagnostic accuracy of FNB needles in patients with atypical AIP. The development of new devices may increase the clinical management of AIP.

## Figures and Tables

**Figure 1 diagnostics-11-01653-f001:**
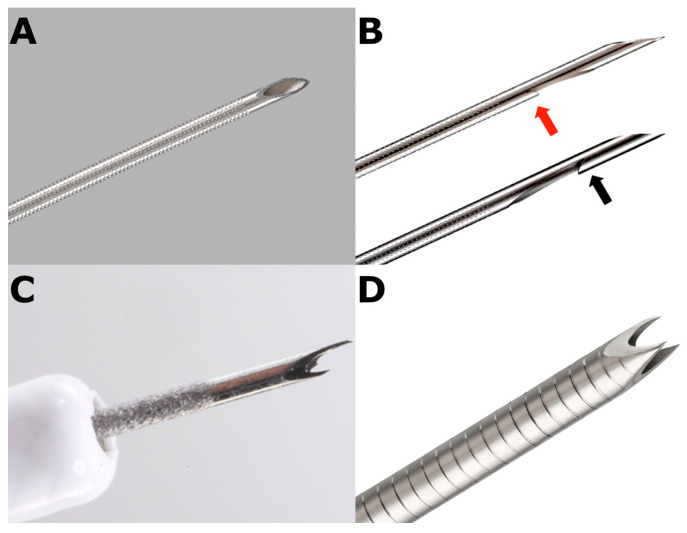
(**A**) the endoscopic-ultrasound fine-needle aspiration (EUS-FNA) needle has a standard “lancet-tip”; (**B**) the two versions of “side-fenestrated” needles (ProCore^TM^) for EUS-guided fine needle biopsy (FNB): the “old” one has a “lancet-tip” and a reverse-bevel at the distal end of a lateral fenestration (black arrow); the new one has a Menghini type tip and a straight-bevel located on the proximal side of the lateral window (red arrow); (**C**) the “fork-tip” needle (SharkCore^TM^) for EUS-guided fine needle biopsy (FNB) has six distal cutting-edge surfaces. A prominent, longer tip edge improves tissue access and an opposite shorter tip edge helps capture tissue and drive it into the lumen; (**D**) the “crown-tip” needle (Acquire^TM^) for EUS-guided fine needle biopsy (FNB) has three symmetric point cutting surfaces to provide stability at the tip and enhanced penetration.

**Figure 2 diagnostics-11-01653-f002:**
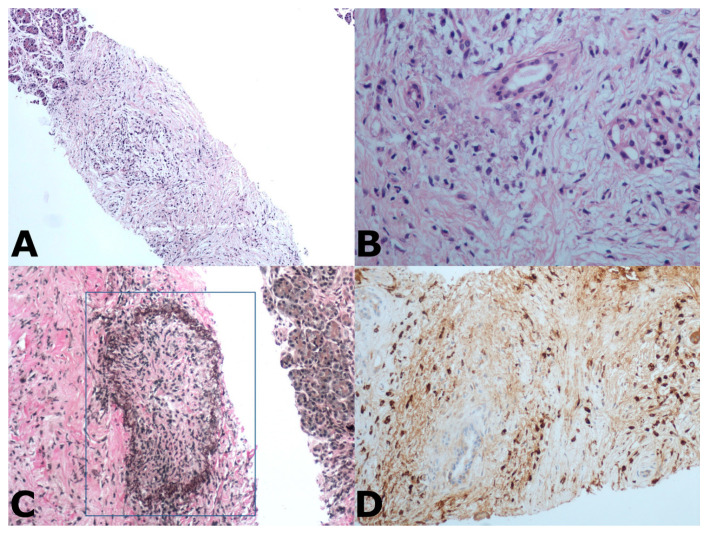
(**A**) Endoscopic ultrasound fine-needle biopsy (EUS-FNB) of the pancreas reveals an exuberant proliferation of fibroblasts and inflammatory cells replacing the normal exocrine gland, that is still recognizable at the left top of the tissue fragment; (**B**) At higher magnification, the storiform fibrosis, composed by fusiform fibroblasts with elongated cytoplasm, and the intermingled plasma cells surround two small intra-acinar ducts, that are the only vestigial of the pre-existing acinar architecture; (**C**) Obliterative phlebitis is highlighted by Van Geison staining. The elastic fibers (in black) surround the vascular lumen that is obliterated by a fibro-inflammatory tissue (blue square); (**D**) IgG4 positive plasma cells (dark points) are recognizable in the connective tissue around a residual small duct (in the middle).

**Table 1 diagnostics-11-01653-t001:** Studies focused on EUS-guided FNA using a 22-gauge needle in type 1 AIP. Histological findings and histological criteria levels are based on ICDC (level 1 indicates more than two cardinal histological criteria; level 2 indicates two cardinal histological criteria). Nr. (%); AIP, autoimmune pancreatitis.

	Imai 2011	Ishikawa 2012	Kanno 2012	Kanno 2016	Morishima 2016	Cao 2018	All
Number of Patients	21	47	25	78	50	27	248
Plasma cell infiltration	0 (0)	16 (34)	23 (92)	43 (55)	36 (72)	18 (67)	136 (55)
IgG4 + plasma cells	0 (0)	10 (21)	9 (36)	19 (24)	27 (54)	8 (30)	73 (29)
Storiform fibrosis	0 (0)	34 (72)	20 (80)	49 (63)	0 (0)	18 (67)	121 (49)
Obliterative phlebitis	0 (0)	0 (0)	4 (16)	38 (49)	0 (0)	0 (0)	42 (17)
Level 1 for type 1 AIP	0 (0)	9 (19)	14 (56)	32 (41)	0 (0)	5 (19)	60 (24)
Level 2 for type 1 AIP	0 (0)	5 (11)	6 (24)	13 (17)	27 (54)	12 (44)	63 (25)
Level 1 or 2 for type 1 AIP	0 (0)	14 (30)	20 (80)	45 (58)	27 (54)	17 (63)	123 (50)

**Table 2 diagnostics-11-01653-t002:** Results of EUS-guided FNB in type 1 AIP. Histological findings and histological criteria levels are based on ICDC (level 1 indicates more than 2 cardinal histological findings; level 2 indicates 2 cardinal histological findings). Nr. (%); G, gauge; AIP, autoimmune pancreatitis.

	Kurita 2020	Kurita 2020	Ishikawa 2020	Notohara 2020	Oppong 2020	Oppong 2020	Tsutsumi 2021	All
Number of Patients	50	51	56	85	6	18	14	280
Needle Type	Franseen	Forward bevel	Franseen	Not specified	Reverse bevel	Fork-tip	Menghini type	/
Needle diameter	22-G	20-G	22-G	19, 20, 22-G	22, 20, 19-G	25, 22-G	21-G	/
Plasma cell infiltration	42 (84)	31 (61)	55 (100)	19 (22)	0 (0)	12 (67)	12 (86)	171 (61)
IgG4 + plasma cells	38 (76)	22 (43)	36 (65)	73 (86)	1 (17)	14 (78)	9 (64)	193 (69)
Storiform fibrosis	28 (56)	13 (25)	40 (73)	32 (38)	0 (0)	11 (61)	5 (36)	129 (46)
Obliterative phlebitis	12 (24)	7 (14)	24 (44)	24 (28)	0 (0)	8 (44)	1 (7)	76 (27)
Level 1 for type 1 AIP	28 (56)	13 (26)	32 (58)	22 (26)	0 (0)	13 (72)	5 (36)	113 (40)
Level 2 for type 1 AIP	11 (22)	10 (20)	19 (34)	23 (27)	0 (0)	1 (7)	4 (29)	68 (24)
Level 1 or 2 for type 1 AIP	39 (78)	23 (45)	51 (93)	45 (53)	0 (0)	14 (78)	9 (64)	181 (65)

**Table 3 diagnostics-11-01653-t003:** Studies including type 2 AIP patients; FNA, fine-needle aspiration; FNB, fine-needle biopsy; NA, not available.

	Kanno 2012	Ishikawa 2012	Ishikawa 2020	Detlefsen 2017	Matsumoto 2021	All
Patients with type 2 AIP included	1	3	1	2	1	8
Type of Tissue Acquisition	FNA	FNA	FNB	FNB	FNA	/
GELs (level 1 for type 2 AIP)	1	0	0	1	1	3 (37)
Granulocytic infiltrate (level 2 for type 2 AIP)	NA	3	1	2	0	6 (75)

## Data Availability

Not applicable.

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
