# Peer review of "The Role of EUS-Guided FNA and FNB in Autoimmune Pancreatitis"

_diagnostics, 2021, doi:10.3390/diagnostics11091653_

Round 1
Reviewer 1 Report
The authors provided a comprehensive and very interesting summary of EUS-FNA/FNB, a technique for tissue sampling in autoimmune pancreatitis. I would however like to ask the authors to consider my suggestion to make this paper more understandable.
#1 Please clarify the definitions of EUS-FNA and FNB. The authors describe them in Section 2, but the expression seems to imply cytology, which is inadequate and confusing. In many papers on AIP, the term EUS-FNA seems to refer to biopsy with a conventional straight needle (EUS-FNA needle).
#2 For the reader's understanding, some images would be helpful; provide the histopathological images characteristic of AIP (actual samples collected by FNA/FNB) and the apical structure of the FNB needle, if possible.
#3 The guidelines of each academic society should also be mentioned.
Kawa S, Kamisawa T, Notohara K, et al. Japanese clinical diagnostic criteria for autoimmune pancreatitis, 2018: revision of Japanese clinical diagnostic criteria for autoimmune pancreatitis, 2011. Pancreas 2020; 49: e13–e14.
Lohr J, Beuers U, Vujasinovic M, et al. European Guideline on IgG4-related digestive disease – UEG and SGF evidence-based recommendations. United European Gastroenterol J 2020; 8: 637–666.
Author Response
# 1 Please clarify the definitions of EUS-FNA and FNB. The authors describe them in Section 2 but the expression seems to imply cytology, which is inadequate and confusing. In many papers on AIP, the term EUS-FNA seems to refer to biopsy with conventional straight needle (EUS-FNA needle).
Answer: Thank you very much for this comment. We modified the sentences as follows to clarify better the the term FNA: “The aim of FNA is the aspiration of cells obtained from a target tissue using a conventional straight needle. This technique allows the pathologist to evaluate the presence and features of abnormal cells in the aspirated sample and sometimes of small bioptic samples”.
# 2 For the readers understanding, some images would be helpful; provide histopathological images characteristic of AIP (actual samples collected by FNA/FNB) and the apical structure of the FNB needle, if possible.
Answer: We added histopathological image of FNB and of the apical structure of the different FNB needles.
# 3 The guidelines of each academic society should also be mentioned.
Answer: We completely agree with this comment. The requested papers have been added to the references.
Reviewer 2 Report
Very good literature review and FNA/FNB comparison., with important conclusions
Author Response
Very good literature review and FNA/FNB comparison, with important conclusions.
Answer: Thank you very much for the appreciation
Reviewer 3 Report
I would recommend -
- EUS description of AIP and add related image as an example
- Adding histological images (optional)
Otherwise this is an excellent review. The conclusion paragraph needs to be in one single paragraph (and not in multiple sentences).
Author Response
# 1 EUS description of AIP and add related image as an example.
Answer: Thank you very much for this comment. The following sentence has been added to describe AIP appearance at EUS: “Endoscopic ultrasound is an excellent technique for the morphologic evaluation of the pancreas, mainly characterized by altered echotexture, reduced echogenicity and hyperechoic strands, but in focal forms of AIP, has non-specific appearance. Therefore, tissue acquisition is required for a definitive diagnosis.”
# 2 Adding histological images (optional).
Answer: Thank you very much for this comment. We agree that the presence of a histopathological image could be helpful for the reader. An image of pancreatic FNB has been added.